# HDL and LDL: Potential New Players in Breast Cancer Development

**DOI:** 10.3390/jcm8060853

**Published:** 2019-06-14

**Authors:** Lídia Cedó, Srinivasa T. Reddy, Eugènia Mato, Francisco Blanco-Vaca, Joan Carles Escolà-Gil

**Affiliations:** 1Institut d’Investigacions Biomèdiques (IIB) Sant Pau, Sant Quintí 77, 08041 Barcelona, Spain; lcedo@santpau.cat (L.C.); emato@santpau.cat (E.M.); 2CIBER de Diabetes y Enfermedades Metabólicas Asociadas (CIBERDEM), Monforte de Lemos 3-5, 28029 Madrid, Spain; 3Department of Molecular and Medical Pharmacology, David Geffen School of Medicine, University of California, Los Angeles, CA 90095-1736, USA; sreddy@mednet.ucla.edu; 4Departament de Bioquímica i Biologia Molecular, Universitat Autònoma de Barcelona, Av. de Can Domènech 737, 08193 Cerdanyola del Vallès, Spain; 5CIBER de Bioingeniería, Biomateriales y Nanomedicina (CIBER-BBN), Monforte de Lemos 3-5, 28029 Madrid, Spain

**Keywords:** Breast cancer, cholesterol, 27-hydroxycholesterol, HDL, LDL, cholesterol-lowering therapies

## Abstract

Breast cancer is the most prevalent cancer and primary cause of cancer-related mortality in women. The identification of risk factors can improve prevention of cancer, and obesity and hypercholesterolemia represent potentially modifiable breast cancer risk factors. In the present work, we review the progress to date in research on the potential role of the main cholesterol transporters, low-density and high-density lipoproteins (LDL and HDL), on breast cancer development. Although some studies have failed to find associations between lipoproteins and breast cancer, some large clinical studies have demonstrated a direct association between LDL cholesterol levels and breast cancer risk and an inverse association between HDL cholesterol and breast cancer risk. Research in breast cancer cells and experimental mouse models of breast cancer have demonstrated an important role for cholesterol and its transporters in breast cancer development. Instead of cholesterol, the cholesterol metabolite 27-hydroxycholesterol induces the proliferation of estrogen receptor-positive breast cancer cells and facilitates metastasis. Oxidative modification of the lipoproteins and HDL glycation activate different inflammation-related pathways, thereby enhancing cell proliferation and migration and inhibiting apoptosis. Cholesterol-lowering drugs and apolipoprotein A-I mimetics have emerged as potential therapeutic agents to prevent the deleterious effects of high cholesterol in breast cancer.

## 1. Introduction

Breast cancer is the third most common cancer overall, with an estimated incidence of 1.7 million cases in 2016 and a 29% increase in incident cases between 2006 and 2016. Moreover, breast cancer was the fifth leading cause of cancer deaths for both sexes in 2016 and the primary cause of death for women [1]. A substantial proportion of the worldwide burden of cancer could be prevented; however, improved primary prevention of cancer requires identification of risk markers [2]. Reproductive, hormonal factors, and unhealthy lifestyles that trigger obesity are considered significant risk factors for breast cancer [3]. Obesity represents a potentially modifiable risk factor that could increase the risk of breast cancer in women [4,5]. The biological association between obesity and disease risk, at least in part, may be related to circulating lipid levels and tissue lipid metabolism [6].

Cancer cells show specific alterations in different aspects of lipid metabolism, which can affect the availability of structural lipids for the synthesis of membranes, contribution of lipids to energy homeostasis, and lipid signaling functions, including the activation of inflammation-related pathways. All these changes are related to important cellular processes, including cell growth, proliferation, differentiation, and motility [7]. The interplay among cholesterol, lipoproteins, proinflammatory signaling pathways, and tumor development has mainly been studied in breast cancer cells and experimental models in vivo. Furthermore, in humans, both benign and malignant proliferation of breast tissue were associated with changes in plasma lipid and lipoprotein levels [8], despite that epidemiological data on the association between lipoproteins and breast cancer showed inconclusive results [9,10,11]. This article reviews the progress to date in research on the role of cholesterol and its main lipoprotein transporters, the low-density and high-density lipoproteins (LDL and HDL), on breast cancer development, mainly focusing on recent findings in human trials and those obtained in experimental models of breast cancer. PubMed was searched comprehensively with combinations of the keyword Breast Cancer and the rest of keywords related with cholesterol and lipoproteins.

## 2. Association of Cholesterol in Breast Cancer Risk: Clinical and Epidemiological Studies

Study of the relationship between serum cholesterol levels and risk of cancer is of special interest and has sparked debate, especially with the expansion of lipid-modifying therapies and more aggressive cholesterol goals to reduce the risk of cardiovascular events [12]. However, different studies have produced divergent results. Indeed, one study found that total cholesterol was associated with the risk of breast cancer [13], but others failed in finding such an association [14,15,16,17,18], or they even found that total cholesterol was inversely associated with the risk of breast cancer [19].

Since cholesterol is mainly transported by LDL and HDL, several clinical trials have associated them with breast cancer. A clinical study in which the lipid profile was assessed in women with breast cancer showed that LDL cholesterol (LDL-C) levels at diagnosis was a prognostic factor of breast tumor progression. A systemic LDL-C level above 117 mg dL^−1^ was found to be a predictive factor of tumor stage, and it was positively associated with worse prognosis because of a higher histological grade, higher proliferative rate, and more advanced clinical stage [20] (Table 1). Moreover, patients with LDL-C above 144 mg dL^−1^ were also prone to have lymph node metastasis [20]. More importantly, a Mendelian randomization study found that genetically raised LDL-C was associated with a higher risk of breast cancer [11]. However, other meta-analyses and prospective studies found no association between LDL-C and breast cancer risk [9,10,16,21]; some trials even found that LDL-C or non-HDL were inversely associated with the risk of breast cancer [14,22] (Table 1).

Concerning HDL-C, discordant results were also found. One prospective study with a follow-up time of 11.5 years found an inverse association between HDL-C and breast cancer risk [19], and retrospectively collected clinical data showed that decreased HDL-C levels had a significant association with worse overall survival in breast cancer patients [23] (Table 1). In contrast, a Mendelian randomization study showed that raised HDL-C increased the risk of estrogen receptor (ER)-positive breast cancer [11] (Table 1). It should also be noted that other studies failed to find any association between HDL-C and breast cancer risk [10,21,24] or survival [24]. Moreover, controversy also exists when considering the menopausal status of patients (Table 1). Some studies have found that low HDL-C among premenopausal women increased breast cancer risk [9,25,26], while others found that low HDL-C was associated with an increased postmenopausal risk of breast cancer [16,27].

In summary, although some studies failed to find associations between lipoproteins and breast cancer, the results of some large clinical trials seem to point to a direct association between LDL-C and breast cancer risk as well as an inverse association between HDL-C and breast cancer risk. It is important to note that clinical or methodological differences in the design of the studies, including variation in geographic regions, menopausal status, number of cases, or follow-up length, could explain the discrepancies found in these studies (summarized in Table 1). For this reason, basic scientific research can contribute to determining potential underlying mechanisms that may explain these associations [12].

## 3. Hypercholesterolemia and Breast Cancer

Diet and obesity are important risk factors for breast cancer development [5,32]. High cholesterol intake was found to be positively associated with the risk of breast cancer, mainly among postmenopausal women [33,34]. To address interactions between body weight and dietary fat intake on subsequent mammary tumor development, a study was performed in which female murine mammary tumor virus (MMTV)-transforming growth factor α (TGFα) mice consumed a moderately high-fat diet [35]. The MMTV promoter specifically directs expression to the mammary epithelium [36], obtaining a model that recapitulates human breast cancer progression from early hyperplasia to malignant breast carcinoma [37]. These mice exhibited mammary tumor latency inversely related to their body fat, suggesting that body fat may be the mediating factor of the effect of a high-fat diet on mammary tumor development [35]. Moreover, the expression of a number of proteins associated with leptin and apoptosis signaling pathways were also affected by diet in the mammary tumors of these animals [38].

Some studies have specifically addressed the role of dietary cholesterol in the regulation of tumor progression in different experimental mouse models of breast cancer. Llaverias et al. studied the role of a high-fat/high-cholesterol (HFHC) diet administration in MMTV polyoma middle T (PyMT) oncogene transgenic mice and found that the HFHC diet accelerated and enhanced tumor progression in these mice [39]. Plasma cholesterol levels were reduced during tumor development but not prior to its initiation, providing new evidence for an increased utilization of cholesterol by tumors and for its role in tumor formation [39]. Another group administered an HFHC diet to female immunodeficient mice implanted orthotopically with MDA-MB-231 cells and found that diet induced angiogenesis and accelerated breast tumor growth in this model of breast cancer [40]. 

The role of dyslipidemia in breast cancer growth and metastasis was also explored in hypercholesterolemic apolipoprotein E knockout mice (apoE^–/–^) fed an HFHC diet and injected with non-metastatic Met-1 and metastatic Mvt-1 mammary cancer cells derived from PyMT mice and c-Myc/vegf tumor explants, respectively [41]. The apoE glycoprotein is a structural component of all lipoprotein particles other than LDL, and it acts as a ligand of lipoprotein receptors and participates in the uptake of lipids into cells. The absence of apoE leads to the accumulation of cholesterol and triglycerides in plasma [42]. ApoE^–/–^ mice exhibited increased tumor growth and displayed a greater number of spontaneous metastases to the lungs. The results in tumor growth were only observed when an HFHC diet was administered to the mice, not when they were fed a standard chow diet [41]. Therefore, although the uptake of cholesterol via apoE was blocked, other adipocyte apoE-independent receptors, such as LDL receptor (LDLR) [43], may be involved in the cholesterol uptake by cancer cells. Moreover, the phosphoinositide 3-kinase (PI3K)/Akt pathway, involved in proinflammatory and cell proliferation signals, was found to be one mediator of the tumor-promoting activity of hypercholesterolemia [41].

Whereas the selection of HFHC diets for these studies reflects current dietary trends, this approach has not allowed an evaluation of the specific effect of cholesterol on tumor biology [44]. To directly address this question, PyMT mice were administered a high-cholesterol diet from weaning and developed palpable tumors earlier than mice on a control chow diet were [45]. High-cholesterol diet administration to mice injected with different breast cancer cell lines (human breast cancer HTB20 and MDA-MB-231, and the mouse breast cancer cell line 4 T1) also promoted breast tumor growth. Tumors of animals in the high-cholesterol diet group showed a higher proliferative ratio than those from chow-fed mice, and lung metastasis was increased [46].

## 4. 27-Hydroxycholesterol and Breast Cancer

Estrogen receptor α-induced signal transduction controls the growth of most breast cancers [47]. 27-hydroxycholesterol (27-HC), one of the most prevalent oxysterols, was identified as an endogenous selective ER modulator (SERM) and liver X receptor (LXR) agonist [48]. This oxysterol is generated enzymatically from cholesterol by the P450 enzyme sterol 27-hydroxylase CYP27A1. CYP27A1 is abundant in the liver, but it is also expressed in the intestine, vasculature, brain, and macrophages. 27-HC is mainly transported in association with HDL and LDL, primarily in the esterified form [49]. Regarding its catabolism, 27-HC is hydroxylated by oxysterol 7α-hydroxylase CYP7B1, which is also abundant in the liver [50]. 

The first evidence for 27-HC’s role in breast cancer began with studies that found that it stimulated the growth of ER-positive MCF-7 cells but not that of ER-negative MCF-10 cells. The effect of a concentration of 1–2 µM of 27-HC was similar to that of 1–2 nM of 17β-estradiol [51]. The proliferative role of 27-HC in vitro on MCF-7 cells was also confirmed by others, who also reported that 27-HC increased tumor growth in vivo in PyMT mice and in murine or human cancer cell xenografts [45,52]. 27-HC was also found to hasten metastasis to the lungs, an effect that implicated LXR activation [45]. 27-HC also hastened myeloid immune cell functions, as it was found that this oxysterol increased the number of polymorphonuclear neutrophils and γδ T cells as well as decreased cytotoxic CD8^+^ T cells within tumors and metastatic lesions [53].

In human breast cancer tissue, 27-HC concentration was found to increase because of decreased catabolism, since *CYP7B1* gene expression was downregulated, whereas *CYP27A1* remained unchanged. Moreover, increased *CYP7B1* mRNA was correlated with better survival [52]. Consistently, Nelson et al. found increased CYP27A1 protein expression in higher grade tumors [45]. Nevertheless, the first prospective epidemiological study on prediagnosis of circulating 27-HC and breast cancer risk showed an inverse association between blood 27-HC and breast cancer risk among postmenopausal women. The authors hypothesized that 27-HC-associated inhibition of estradiol–ER binding outweighed 27-HC’s agonistic effect in human breast cancer [54].

Unlike humans, mice do not normally become severely hypercholesterolemic when fed an HFHC diet [44]. To circumvent this limitation, breast cancer cells were implanted in mice in which the mouse *Apoe* gene was replaced with the human *APOE3* allele, which codes for the most frequent human isoform. The animals on an HFHC diet exhibited both increased cholesterol and 27-HC in plasma as well as promotion of larger tumors, effects that were partially reversed by treatment with the CYP27A1 inhibitor GW273297X [45]. 

Several studies investigated the potential mechanisms involved in 27-HC-induced breast cancer development. First, 27-HC inhibited p53 protein and activity in MCF-7 cells via ER. The oxysterol increased p53 regulator mouse double minute 2 (MDM2) levels and enhanced interaction between p53 and MDM2, suggesting that 27-HC proliferation depended on MDM2-mediated p53 degradation. Interestingly, estradiol, the main physiological endogenous ligand for ER, which had similar effects to 27-HC on cell proliferation, had no effect on p53 activity; this demonstrates that 27-HC may contribute to ER-positive breast cancer progression via different mechanisms compared with known estrogens [55]. Another study found that 27-HC increased Myc protein stability (a critical oncogene that can promote proliferation, migration, and invasion of cancer cells) by reducing its dephosphorylation and ubiquitination for proteasomal degradation [56]. Signal transducer and activator of transcription (STAT)-3 is an important transcription factor that can target c-Myc, vascular endothelial growth factor (VEGF), cyclin D1, matrix metalloproteinase (MMP) 2, and MMP9 to promote the development of cancer involving tumor proliferation, invasion, metastasis, and angiogenesis [57]. 27-hydroxycholesterol induced activation of STAT-3, which promoted the angiogenesis of breast cancer cells via proinflammatory-related reactive oxygen species (ROS)/STAT-3/VEGF signaling [58]. Moreover, it induced the epithelial–mesenchymal transition (EMT) [59], a mechanism that promotes migration and invasion, via STAT-3/MMP9 and STAT-3/EMT [60], in both ER-positive and ER-negative breast cancer cells. Furthermore, 27-HC causes greater macrophage infiltration and exacerbation of inflammation in the setting of hypercholesterolemia [61], thereby providing a link between inflammation and cancer development. Collectively, mechanisms involved in 27-HC-promoted progression of breast cancer are complex. Therefore, seeking effective measures to prevent 27-HC-caused pathogenicity is difficult, and further studies should be carried out with an emphasis on deeply investigating the potential mechanisms involved in 27-HC breast cancer promotion [58].

The discovery of 27-HC as an endogenous ER ligand that promotes ER-positive breast tumor growth could help explain why some breast cancer patients are resistant to aromatase inhibitors [62]. In this way, 27-HC may act as an alternate estrogenic ligand in a low-estrogen environment [63]. Assessments of 27-HC or their metabolic enzymes’ abundance in tumors could aid in personalizing hormone-based therapy [64]. 

## 5. Low-Density Lipoprotein and Breast Cancer

Proliferating cancer cells have an increased cholesterol need. Increased LDLR expression was demonstrated in breast cancer tissue to increase the uptake of LDL-C from the bloodstream [65]. In vitro, LDLR gene and protein expression was found increased in ER-negative MDA-MB-231 cells in contrast to ER-positive MCF-7 cells [66,67]. Accordingly, LDL-C mainly promoted proliferation [68,69,70] and migration [46,71] in ER-negative cells, but this was not evident in ER-positive cell lines. This difference between the two cell types corresponded to a greater ability of ER-negative cells to take up, store, and utilize exogenous cholesterol because of the increased activity of acyl-CoA:cholesterol acyltransferase 1 (ACAT1) [68]. The Women’s Intervention Nutrition Study (WINS) found that a low-fat diet mainly extended relapse-free survival in women with ER-negative breast cancer [72]. At least in part, that ER-negative breast cancer cells differentially uptake and store cholesterol may explain the differential effect of a low-fat diet on human breast cancer recurrence [68]. Another study found that LDL-C also induced proliferation in ER-positive BT-474 breast cancer cells [46]. This discrepancy could be because BT-474 cells usually express the Her2 (ErbB2) receptor [73]; furthermore, high plasma LDL-C levels were found to be associated with Her2-positive breast cells [20]. It is noteworthy that the Her2-positive and triple-negative subtypes are the most aggressive breast cancers [74].

Beyond in vitro studies, tumors from breast cancer cells with high LDLR expression (murine MCNeuA (Her2-positive) and human MDA-MB-231 (triple-negative), respectively) have been incrementally grown in immunocompetent (LDLR^–/–^ and apoE^–/–^) and immunodeficient (Rag1^–/–^/LDLR^–/–^ and Rag1^–/–^/apoE^–/–^) mouse models of hyperlipidemia with increasing serum LDL concentrations. Importantly, silencing LDLR in the tumor cells reduced tumor growth [67]. 

Finally, in human samples, *LDLR* and *ACAT1* were also found to be increased in Her2-positive and triple-negative tumors compared with luminal A tumors. Her2-positive and triple-negative tumors were more cholesteryl ester-rich and had higher histological grades, Ki-67 expression, and tumor necrosis. Therefore, cholesteryl ester accumulation due to increased LDL-C internalization and esterification was associated with breast cancer proliferation [75]. In line with these findings, higher LDLR expression was found to be associated with a worse prognosis in patients who underwent systemic therapy [67]. Overall, elevated circulating LDL and breast cancer expression of LDLR have roles, at least in Her2-positive and triple-negative breast cancers, in disease progression and disease-free survival.

### Oxidized Low-Density Lipoprotein and Breast Cancer

Lipid peroxidation is associated with carcinogenesis [76]. Lipid peroxidation metabolites cause structural alterations in DNA and decrease DNA repair capacity through their direct interaction with repair enzymes [77]. The oxidation of LDL affects both protein and lipid contents, resulting in the formation of peroxidation metabolites. Patients with breast cancer exhibited elevated serum levels of oxidized LDL (oxLDL) [78]. Moreover, serum oxLDL levels were associated with increased breast cancer risk [78]. Oxidized LDL was also reported to trigger pro-oncogenic signaling in MCF10A cells; concretely, cells treated with oxLDL showed a dose-dependent stimulation of proliferation mediated by stimulation of the microRNA miR-21, which, in turn, activated the related proinflammatory PI3K/Akt signaling pathways [79]. 

OxLDL lecithin-like receptor 1 (OLR1) is the main receptor for internalization of oxLDL. It is overexpressed in human breast cancer and positively correlates to tumor stage and grade [80]. A microarray analysis of hearts of *Olr1* KO mice compared with wild-type mice showed a reduction in the expression of nuclear factor κB (NF-κB) target genes involved in cellular transformation (regulation of apoptosis, proliferation, wound healing, defense response, immune response, and cell migration) as well as an inhibition of key enzymes involved in lipogenesis. The human breast cancer cell line HCC1143 showed increased *OLR1* expression compared with the normal mammary epithelial cell line MCF10A [81]. Forced overexpression of *OLR1* in both cell lines resulted in upregulation of NF-κB and its target pro-oncogenes involved in the inhibition of apoptosis (*BCL2*, *BCL2A1*, and *TNFAIP3*) and regulation of the cell cycle (*CCND2*) in HCC1143 cells. Moreover, upregulation of *OLR1* in breast cancer cell lines enhanced cell migration [81,82]. In line with these findings, *OLR1* depletion by siRNAs, or ORL1 inhibition by antibodies or a recombinant OLR1 protein, significantly suppressed the invasion and migration of breast cancer cells [81,82,83]. *TBC1D3* is a hominoid-specific oncogene that also regulates migration of human breast cancer cells. *TBC1D3* was found to stimulate the expression of *OLR1*, and this *TBC1D3*-induced *OLR1* expression was regulated by tumor necrosis factor α (TNFα)/NF-κB signaling [84]. Therefore, *OLR1* may function in special situations, such as obesity and chronic inflammation, to increase breast cancer susceptibility. 

## 6. High-Density Lipoprotein and Breast Cancer

Controversy exists about the association between HDL-C levels and breast cancer risk, as detailed in Section 2. In the present section, experimental data evaluating the role of HDL in breast cancer development are reviewed. In vitro analyses have shown that HDL stimulated proliferation in both ER-positive [69,85] and ER-negative breast cancer cell lines [69] in a dose-dependent manner, but ER-negative cells showed a higher response [69]. Human HDL3 also induced migration and activated Akt and extracellular signal-regulated kinases (ERK)1/2 signal transduction pathways in both MCF7 and MDA-MB-231 cells [86].

The scavenger receptor class B type I (SR-BI) acts as an HDL receptor and mediates its cholesterol uptake in breast cancer cells [87]. The receptor SR-BI is abundantly expressed in human breast cancer tissue compared with adjacent normal tissue [88]. Moreover, high SR-BI expression was found related to tumor aggressiveness and poor prognosis in breast cancer [75,89,90], whereas knockdown of SR-BI in vitro attenuated Akt activation and inhibited breast cancer cell proliferation and migration [86]. Moreover, HDL-induced proliferation was blocked in transfected MCF-7 cells with a mutant, nonfunctional SR-BI [88]. Beyond in vitro studies, mice injected with SR-BI-knockdown breast cancer cells showed a decreased tumor burden, accompanied with reduced Akt and ERK1/2 activation, reduced angiogenesis, and increased apoptosis [86]. Therefore, cholesteryl ester entry via HDL-SR-BI and Akt signaling seems to play a critical role in the regulation of cellular proliferation and migration and tumor growth. SR-BI was also found to increase concomitantly with an increased number and size of tumors in PyMT mice fed an HFHC diet compared with those fed a chow diet. However, cholesterol was not found accumulated in the mammary tumors, suggesting that even if tumor cholesterol uptake was increased, cholesterol was probably metabolized to sustain a high level of proliferation [39].

Serum HDL particles contain either a single copy or multiple copies of apolipoprotein A-I (apoA-I), the most abundant HDL apolipoprotein [91]. Apolipoprotein A-I plays a role in promoting cholesterol release from cells; possesses anti-inflammatory, antioxidant, and antiapoptotic properties; and influences innate immunity [92]. The levels of apoA-I have normally been found to be inversely associated with breast cancer risk [19,93,94], although one study found that apoA-I was positively associated with breast cancer [21]. Our group showed that human apoA-I-containing HDL could not hinder breast tumor development in PyMT mice. While overexpression of human apoA-I reduced the levels of oxLDL, 27-HC levels were increased, which could promote tumor growth [95]. Concerning apoA-II, the second major protein constituent of HDL [96], our research group showed that human apoA-II-containing HDL increased the breast tumor burden in PyMT mice (Figure 1A) (unpublished results). These results may be related with the apoA-II-mediated alteration in HDL remodeling, decreased capacity to protect against LDL oxidative modification and its proinflammatory actions, and postprandial hyperlipidemia (Figure 1B) [97,98].

### Dysfunctional High-Density Lipoprotein and Breast Cancer

Under conditions of oxidative stress, HDL can be oxidatively modified, and these modifications may have an effect on HDL function. Hypochlorite-oxidized HDL was found to stimulate cell proliferation, migration, invasion, and adhesion in vitro, involving the protein kinase C (PKC) pathway, which regulates numerous cellular responses including cell proliferation and the inflammatory response. This modified HDL promoted breast cancer cell pulmonary and hepatic metastasis compared with normal HDL in vivo. Interestingly, in this study, normal HDL reduced the metastasis of MCF7 cells in the liver compared with control animals in which HDL was not injected [99].

In patients with type 2 diabetes mellitus (T2DM), HDL can be modified into dysfunctional glycated HDL and oxidized HDL [100]. Indeed, T2DM patients have a 20% increased risk of breast cancer compared with nondiabetic subjects [101]. In this context, diabetic HDL was found to have a stronger capability to promote cell proliferation, migration, and invasion of breast cancer cells through the Akt, ERK, and p38 mitogen-activated protein kinase (MAPK) pathways. These observations were also found in glycated and oxidized HDL produced in vitro, compared with normal HDL [102]. Pretreatment with diabetic, glycated, and oxidized HDL also promoted the metastasis capacity of breast cancer cells in vivo, and it increased their capacity of adhesion to human umbilical vein endothelial cells (HUVECs) and attachment to the extracellular matrix in vitro, compared with normal HDL. These effects mainly were due to elevated PCK activity, which, in turn, could stimulate secretion of integrins, which are important in promoting breast cancer metastasis [103]. Similarly, HDL isolated from patients with breast cancer complicated with T2DM promoted an increase in breast cancer cell adhesion to HUVECs and stimulated higher intercellular adhesion molecule 1 (ICAM-1) and vascular cell adhesion molecule 1 (VCAM-1) expression on the cell surfaces of breast cancer cells and HUVECs, along with the activation of PKC, compared with HDL isolated from breast cancer patients. However, in breast cancer patients complicated with T2DM, a lower expression of ICAM-I and VCAM-I was found in their tumor tissue, which may contribute to the metastasis of tumor cells [104]. Collectively, associations between T2DM and breast cancer could be attributed, in part, to alterations in HDL structure and composition and their proinflammatory actions.

## 7. Effects of Cholesterol-Lowering Therapies on Breast Cancer

The studies reviewed indicate that cholesterol and its main metabolite, 27-HC, may increase breast cancer development and metastasis. To address this, cholesterol-lowering drugs have emerged as potential therapies to reverse the deleterious effects of impaired cholesterol metabolism in breast cancer. 

### 7.1. Statins

Statins are inhibitors of the enzyme hydroxy-methyl-glutaryl-coenzyme A reductase (HMGCR), which catalyzes the conversion of HMG-CoA to mevalonate, the rate-limiting step of cholesterol synthesis [105]. In humans, the effect of statins in cancer prevention and treatment remains controversial (Table 2). The use of lipid-lowering drugs, and more concretely, statins, was found to be associated with a reduced risk of breast cancer in older women [106]. Specifically, the use of lipophilic statins but not hydrophilic statins were found to significantly reduce the risk of breast cancer in Thai women [107]. Conversely, other studies, including a large Mendelian randomization study, failed to find a protective effect of statins against breast cancer risk [108,109,110,111,112,113], or they even found a positive association between long-term use of statins and increased risk of breast cancer [114]. In contrast, treatment with statins seems to have a more important effect in protecting against breast cancer recurrence and death [115,116,117,118,119,120,121,122]. Considering the type of statin, lipophilic statins were mainly found to be associated with a reduced risk of breast cancer recurrence or mortality [123,124,125], although hydrophilic statin use was also found to be associated with improved progression-free survival compared with no statin use in inflammatory breast cancer patients [126]. Taken together, HMGCR inhibitors do not seem to protect against breast cancer development, but statins, and more concretely, lipophilic statins, could be a good strategy for protecting against breast cancer recurrence and death.

Statins also exert antiproliferative and cytotoxic effects on breast cancer cells in vitro by increasing apoptosis, autophagy, and cell cycle arrest [127,128]. However, only lipophilic statins show anticancer activity [129], and the ER-negative phenotype seems to be more sensitive than those that overexpress ER [129,130]. ER-positive cell resistance to statin treatment is associated with high expression of cholesterol biosynthesis genes [131].

In vivo studies have also reported controversial results. Atorvastatin was able to reduce the level of circulating cholesterol, and it attenuated enhanced tumor growth and lung metastasis associated with an HFHC diet in a transgenic model in which the murine *Apoe* gene was replaced with the human *APOE3* allele and injected with ER-positive E0771 murine mammary cancer cells [45,53]. Moreover, simvastatin and fluvastatin treatments were found to inhibit tumor growth in mice inoculated with breast cancer cells [129,132], and fluvastatin was also found to reduce the metastatic burden in a murine breast cancer metastasis model [133]. The mechanisms of action of simvastatin included the inhibition of NF-κB transcription factor, which attenuated expression of antiapoptotic Bcl_XL_ and derepressed expression of the antiproliferative/proapoptotic tumor suppressor PTEN, which reduced the phosphorylation of Akt, resulting in decreased cancer cell proliferation and survival [132]. In contrast, statin treatment failed to reduce plasma cholesterol levels or tumor growth in mice injected with breast cancer cells on an HFHC diet [46] or other models of breast cancer in mice and rats [134]. An explanation for these negative results could be that mice are generally unresponsive to statins [135].

Finally, an interesting study investigated the biological effect of short-term lipophilic fluvastatin exposure on in situ and invasive breast cancer through paired tissue, blood, and imaging-based biomarkers in women with a diagnosis of ductal carcinoma in situ or stage 1 breast cancer. Fluvastatin exposure showed reduced tumor proliferation and increased apoptotic activity in high-grade breast cancer, concomitant with a reduction of cholesterol levels [136]. An upregulation of HMGCR was observed in breast cancer patients after two weeks of atorvastatin treatment, which was interpreted as activation of the negative feedback loop controlling cholesterol synthesis. Moreover, in tumors expressing HMGCR before treatment with atorvastatin, the proliferation marker Ki67 was found to be downregulated. In summary, these results suggested that HMGCR was targeted by statins in breast cancer cells in vivo, and that statins could have antiproliferative effects, mostly in HMGCR-positive breast cancers [137]. Importantly, atorvastatin was also found to decrease serum 27-HC and CYP27A1 expression in tumors of breast cancer patients [138].

### 7.2. Ezetimibe

Ezetimibe is a drug that specifically targets intestinal Niemann-Pick C1-Like 1 (NPC1L1) and mediates the inhibition of intestinal sterol absorption [140]. Few studies have explored the effects of ezetimibe on breast cancer. However, considering that statins may have little effect on plasma cholesterol in mice [135], ezetimibe’s action on breast cancer development is of interest. A study by Pelton et al. investigated the effects of ezetimibe administered in an HFHC diet on breast cancer development in an orthotopic breast tumor model, in which mice were implanted with MDA-MB-231 cells. Ezetimibe was able to reduce tumor volume, proliferation, and angiogenesis and increase apoptosis compared with the HFHC-fed mice, achieving similar results to those in mice fed a low-fat/low cholesterol (LFLC) diet. These results were accompanied with a reduction in circulating cholesterol levels, but intratumoral cholesterol levels remained unchanged [40].

To our knowledge, the effects of ezetimibe treatment on breast cancer risk or mortality have not been studied. Only Kobberø Lauridsen et al. explored the effects of genetic variants of *NPC1L1* (–133A>G and V1296V T>C), mimicking treatment with ezetimibe, on breast cancer risk. These researchers found that *NPC1L1* variants were not associated with the risk of breast cancer [141]. 

### 7.3. Phytosterols

Plant sterols, or phytosterols, lower serum LDL-C levels by reducing intestinal cholesterol absorption [142]. Several in vivo studies have tested the efficacy of dietary phytosterol in breast cancer development. Female severe combined immunodeficiency (SCID) mice supplemented with 2% phytosterols and injected with MDA-MB-231 cells exhibited a reduction in serum cholesterol, accompanied with a reduction in tumor size and metastasis to lymph nodes and lungs [143]. In ovariectomized athymic mice injected with MCF-7 cells, supplementation with β-sitosterol, the most common phytosterol, was also able to reduce tumor size [144]. Furthermore, phytosterol supplementation could decrease both the development of mammary hyperplastic lesions and tumor burden in PyMT mice fed an HFHC diet. This protective effect was not observed in mice fed an LFLC diet. A potential mechanism of action of phytosterol was the prevention of lipoprotein oxidation [145]. 

Numerous experimental in vitro studies showed that phytosterols functioned as anticancer compounds acting on host systems to affect tumor surveillance or on tumors to affect tumor cell biology. Mechanisms affecting the tumors include slowing of cell cycle progression, induction of apoptosis, inhibition of tumor metastasis, altered signal transduction, and activation of angiogenesis. Host influences comprise enhancing immune recognition of cancer, influencing hormonal-dependent growth of endocrine tumors, and altering cholesterol metabolism (reviewed in [146,147]).

### 7.4. Other Therapies

Fibrates are agonists of the peroxisome proliferator-activated receptor α (PPARα), which stimulate the expression of genes involved in fatty acid and lipoprotein metabolism, resulting in a shift from hepatic fat synthesis to fat oxidation. Fibrates are used as therapeutic agents for treating dyslipidemia [148]. A meta-analysis of 17 long-term, randomized, placebo-controlled trials found that fibrates had a neutral effect on breast cancer and other cancer outcomes [149].

The levels of HDL-C and apoA-I are inversely related to cardiovascular risk [150]. The beneficial effects of HDL have largely been attributed to apoA-I, and researchers have sought apoA-I mimetic peptides as therapeutic agents based on physical–chemical and biological properties [151]. To our knowledge, a study from our group was the only one to analyze the effects of apoA-I mimetics on breast cancer. In that study, the apoA-I mimetic peptide D-4F was administered to PyMT female mice, and the treatment significantly increased tumor latency and inhibited the development of tumors. D-4F was unable to reduce the levels of 27-HC in the tumors, but it decreased the plasma levels of oxLDL and prevented the oxLDL-mediated proliferative response in MCF-7 cells, suggesting that D-4F inhibited breast cancer by protecting against LDL oxidative modifications [95].

## 8. Concluding Remarks

Results of some large clinical studies indicate a direct association for LDL-C and an inverse association for HDL-C and breast cancer risk; however, these findings have not been reproduced in all epidemiological studies and are still debated. Basic research studies have determined the important role of cholesterol, especially the 27-HC metabolite, and its transporters in breast cancer development. Both LDL and HDL, and their modified forms (oxLDL and oxidized and glycated HDL), may promote breast cancer via several mechanisms. Investigations in breast cancer cells and experimental models in vivo have demonstrated an interplay among modified lipoproteins, proinflammatory signaling pathways, and breast cancer tumorigenic processes (summarized in Figure 2). Cholesterol can be esterified or metabolized to 27-HC, which has been hypothesized to be responsible for stimulating the proliferation of ER-positive breast cancer cells rather than cholesterol (Figure 2). Oxidized LDL as well as oxidized and glycated HDL induce different OLR1 and SR-BI downstream inflammation-related pathways, thereby inhibiting apoptosis and enhancing cell proliferation and migration. Therefore, considering the important role of cholesterol in breast cancer development, cholesterol-lowering drugs and apoA-I mimetics, which possess antioxidant and anti-inflammatory properties, could emerge as potential therapies for preventing the deleterious effects of high cholesterol in breast cancer. Lipophilic statins seem a good strategy for protecting against breast cancer recurrence and death. However, more studies in humans are necessary to evaluate the role of other therapies, such as ezetimibe, phytosterols or fibrates, on breast cancer risk and prognosis.

## Figures and Tables

**Figure 1 jcm-08-00853-f001:**
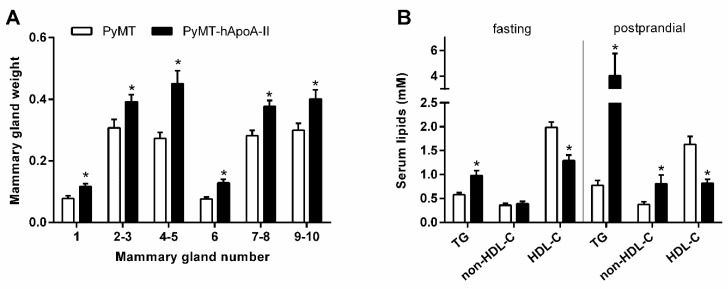
Effects of human apolipoprotein A-II (hApoA-II) overexpression on tumor development in polyoma middle T (PyMT) mice. PyMT mice were backcrossed with hApoA-II transgenic (TG) mice on a C57BL/6 background. The mice were maintained on a regular chow diet until 19 weeks of age, when they were euthanized, and the mammary glands were excised and weighed. Serum lipids were determined after an overnight fasting period and 3 h after a 0.15 mL dose of olive oil by oral gavage. A) Mammary gland weight. B) Serum lipid levels in fasting and postprandial conditions (TG = triglycerides, and HDL-C = high-density lipoprotein cholesterol). Values shown represent the mean ± SEM. A *t*-test was performed to determine the statistical significance between groups. * *p* < 0.05 vs. PyMT mice.

**Figure 2 jcm-08-00853-f002:**
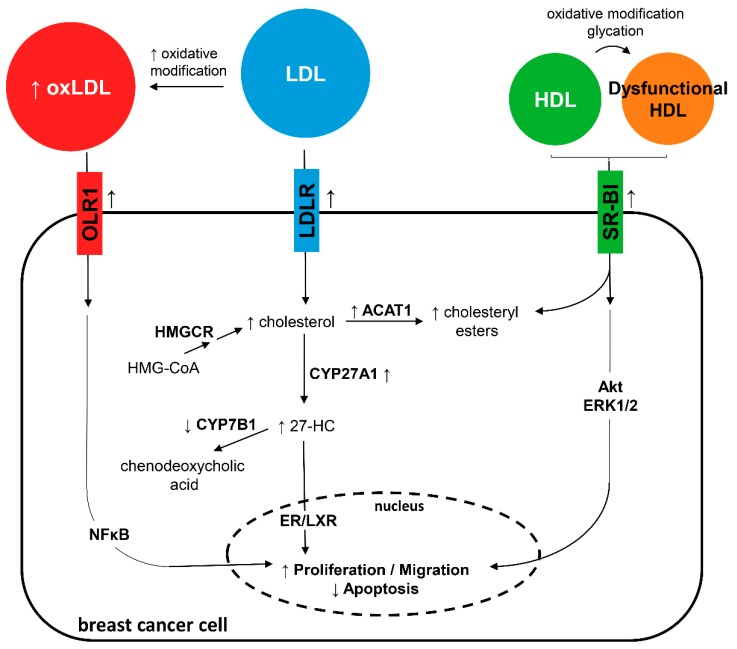
Mechanisms by which low-density lipoprotein (LDL), high-density lipoprotein (HDL), and their modified forms induce proliferation and migration and reduce apoptosis in breast cancer cells. OLR1 = OxLDL lecithin-like receptor 1, LDLR = LDL receptor, SR-BI = scavenger receptor class B type I, HMGCR = hydroxy-methyl-glutaryl-coenzyme A reductase, ACAT1 = acyl-CoA:cholesterol acyltransferase 1, 27-HC = 27-hydroxycholesterol, ERK1/2 = extracellular signal-regulated kinases ½, NFκB = nuclear factor κB, and ER/LXR = estrogen receptor/liver X receptor.

**Table 1 jcm-08-00853-t001:** Clinical and epidemiological studies linking low-density lipoprotein cholesterol (LDL-C) and high-density lipoprotein cholesterol (HDL-C) levels to breast cancer risk.

Reference	Year	Study Design	Participants	Main Findings
Nowak et al. [11]	2018	Mendelian randomization	>400,000	Raised LDL-C increased the risk of breast cancer (OR = 1.09 (1.02–1.18)) and ER-positive breast cancer (OR = 1.14 (1.05–1.24)). Raised HDL-C increased the risk of ER-positive breast cancer (OR = 1.13 (1.01–1.26)).
Ni et al. [16]	2015	Meta-analysis	1,189,635	Inverse association between HDL-C and breast cancer risk among postmenopausal women (RR = 0.45 (0.64–0.93)). No association in premenopausal women. No association between LDL-C and breast cancer risk.
Touvier et al. [9]	2015	Meta-analysis	1,489,484	Inverse association between HDL-C and breast cancer risk among premenopausal women (HR = 0.77 (0.31–0.67)). No association in postmenopausal women. No association between LDL-C and breast cancer risk.
Borgquist et al. [21]	2016	Prospective	5281	No evident associations between LDL-C or HDL-C and breast cancer incidence.
Chandler et al. [10]	2016	Prospective	15,602	No association between LDL-C or HDL-C and breast cancer risk.
His et al. [19]	2014	Prospective	7557	HDL-C was inversely associated with breast cancer risk (HR 1 mmol L^−1^ increment = 0.48 (0.28–0.83)).
Rodrigues dos Santos et al. [20]	2014	Prospective	244	Systemic levels of LDL-C correlated positively with tumor size (Spearman’s r = 0.199, *p* = 0.002).
Kucharska-Newton et al. [25]	2008	Prospective	7575	Modest association of low HDL-C (<50 mg dL^−1^) with breast cancer among premenopausal women (HR = 1.67 (1.06–2.63)). No association in postmenopausal women.
Furberg et al. [27]	2004	Prospective	30,546	The risk of postmenopausal breast cancer was reduced in women in the highest quartile of HDL-C (>1.64 mmol L^−1^) compared with women in the lowest quartile (<1.20 mmol L^−1^; RR = 0.73 (0.55–0.95)). No association was found in premenopausal women.
Li et al. [23]	2017	Retrospective	1044	Decreased HDL-C levels showed significant association with worse overall survival (HR = 0.528 (0.302–0.923)).
Li et al. [28]	2018	Case–control	Total: 3537 Cases: 1054 Controls: 2483	The levels of LDL-C and HDL-C were lower in breast cancer patients than controls (*p* < 0.001).
His et al. [24]	2017	Case–control	Total: 1626 Cases: 583 Controls: 1043	No association between LDL-C or HDL-C and breast cancer risk or survival.
Martin et al. [14]	2015	Case–control	Total: 837 Cases: 279 Controls: 558	HDL-C was positively associated (75th vs. 25th percentile: 23% higher, *p* = 0.05) and non-HDL-C was negatively associated (75th vs. 25th percentile: 19% lower, *p* = 0.03) with breast cancer risk.
Llanos et al. [22]	2012	Case–control	Total: 199 Cases: 97 Controls: 102	Increasing levels of LDL-C were inversely associated with breast cancer risk (OR = 0.41 (0.21–0.81)). Lower levels of HDL-C were associated with a significant increase in breast cancer risk (OR = 1.99 (1.06–3.74)).
Yadav et al. [29]	2012	Case–control	Total: 139 Cases: 69 Controls: 70	Postmenopausal breast cancer patients had higher LDL-C levels (*p* < 0.001) and lower HDL-C levels (*p* = 0.025) than controls. No significant changes in premenopausal women.
Kim et al. [26]	2009	Case–control	Total: 2070 Cases: 690 Controls: 1380	Protective effect of HDL-C on breast cancer was only observed among premenopausal women (OR = 0.49 (0.33–0.72) for HDL-C ≥ 60 vs. <50 mg dL^−1^ (*p* < 0.01)).
Owiredu et al. [30]	2009	Case–control	Total: 200 Cases: 100 Controls: 100	Increased LDL-C levels in postmenopausal breast cancer patients vs. controls (*p* < 0.05). No significant changes in premenopausal women. No changes in HDL-C levels between cases and controls.
Michalaki et al. [31]	2005	Case–control	Total: 100 Cases:56 Controls: 44	A decrease in HDL-cholesterol was observed in patients with breast cancer vs. controls (*p* < 0.05).

LDL-C = low-density lipoprotein cholesterol, HDL-C = high-density lipoprotein cholesterol, ER = estrogen receptor, OR = odds ratio, RR = risk ratio, and HR = hazard ratio. Between brackets, 95% confidence interval.

**Table 2 jcm-08-00853-t002:** Clinical and epidemiological studies linking statin treatment to breast cancer risk.

Reference	Year	Study Design	Participants	Main Findings
Ference et al. [113]	2019	Mendelian randomization	654,783	Genetic inhibition of *HMGCR* did not affect breast cancer risk.
Islam et al. [109]	2017	Meta-analysis	121,399	There was no association between statin use and breast cancer risk.
Liu et al. [123]	2017	Meta-analysis	197,048	Significant protective effects of lipophilic statin use, but not hydrophilic statins, against cancer-specific mortality (HR = 0.57 (0.46–0.70)).
Mansourian et al. [116]	2016	Meta-analysis	124,669	Significant reduction in breast cancer recurrence (OR = 0.792 (0.735–0.853)) and death (OR = 0.849 (0.827–0.870)) among statin users.
Manthravadi et al. [124]	2016	Meta-analysis	75,684	Lipophilic statin use was associated with improved recurrence-free survival (HR = 0.72 (0.59–0.89)).
Wu et al. [119]	2015	Meta-analysis	144,830	There was a significantly negative association between prediagnosis statin use and breast cancer mortality (for overall survival: HR = 0.68 (0.54–0.84), and for disease-specific survival (HR = 0.72 (0.53–0.99)). There was also a significant inverse association between postdiagnosis statin use and breast cancer disease-specific survival (HR = 0.65 (0.43–0.98)). No significant association was detected between statin use and breast cancer risk.
Undela et al. [111]	2012	Meta-analysis	>2.4 million	Statin use and long-term statin use did not significantly affect breast cancer risk.
Bonovas et al. [108]	2005	Meta-analysis	327,238	Statin use did not significantly affect breast cancer risk.
Dale et al. [112]	2005	Meta-analysis	86,936	Statins did not reduce the incidence of breast cancer.
Borgquist et al. [115]	2017	Prospective	8010	Initiation of cholesterol-lowering medication in postmenopausal women with early stage, hormone receptor-positive invasive breast cancer during endocrine therapy was related to improved disease-free survival (HR = 0.79 (0.66–0.95)), breast cancer-free interval (HR = 0.76 (0.60–0.97)), and distant recurrence-free interval (HR = 0.74 (0.56–0.97)).
Murtola et al. [122]	2014	Prospective	31,236	Both postdiagnostic and prediagnostic statin uses were associatedwith a lowered risk of breast cancer death (HR = 0.46 (0.38–0.55) and HR = 0.54 (0.44–0.67), respectively).
Brewer et al. [126]	2013	Prospective	723	Hydrophilic statins were associated with significantly improved progression-free survival compared with no statin (HR = 0.49 (0.28–0.84)) in inflammatory breast cancer patients.
Ahern et al. [125]	2011	Prospective	18,769	Significant reduction in breast cancer recurrence among patients using simvastatin after 10 y of follow up (adjusted HR = 0.70 (0.57–0.86)).
Cauley et al. [106]	2003	Prospective	7528	Older women who used statins had a reduced risk of breast cancer (RR = 0.28 (0.09–0.86), adjusted for age and body weight) compared with nonusers.
Shaitelman et al. [120]	2017	Retrospective	869	Statin use was significantly associated with overall survival (HR = 0.10 (0.01–0.76)) in triple-negative breast cancer.
Smith et al. [121]	2017	Retrospective	6314	Prediagnostic statin use was associated with breast cancer-specific mortality (HR = 0.81 (0.68–0.96)). This reduction was greatest in statin users with ER-positive tumors (HR = 0.69 (0.55–0.85)).
Anothaisintawee et al. [107]	2016	Retrospective	15,718	Using lipophilic statins, but not hydrophilic statins, could significantly reduce the risk of breast cancer (risk difference = –0.0034 (–0.006,–0.001) lipophilic statin users vs. nonusers).
Mc Menamin et al. [139]	2016	Retrospective	15,140	There was no evidence of an association between statin use and breast cancer-specific death.
Sakellaki et al. [117]	2016	Retrospective	610	Statins may be linked to a favorable outcome in early breast cancer patients, especially in younger age groups (HR = 0.58 (0.36–0.94)).
Chae et al. [118]	2011	Retrospective	703	Significant reduction in breast cancer recurrence among patients who used statins (HR = 0.43 (0.26–0.70)). No association was found regarding overall survival.
Schairer et al. [110]	2018	Case–control	Total: 228,973 Cases: 30,004 Controls: 198,969	Statin use did not significantly affect breast cancer risk.
McDougall et al. [114]	2013	Case–control	Total: 2886 Cases: 916 IDC + 1068 ILC Controls: 902	Current users of statins for ≥10 y had increased risk of IDC (OR = 1.83 (1.14–2.93)) and ILC (OR = 1.97 (1.25–3.12)) compared with never users of statins.

OR = odds ratio, RR = risk ratio, HR = hazard ratio, y = years, IDC = invasive ductal carcinoma; and ILC = invasive lobular carcinoma. Between brackets, 95% confidence interval.

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
