# Peer review of "HDL and LDL: Potential New Players in Breast Cancer Development"

_jcm, 2019, doi:10.3390/jcm8060853_

Round 1
Reviewer 1 Report
The authors proposed an interesting review. This review is part of an scientific debate on breast cancer risk. The review has involve a detailed and comprehensive plan. The main subject is well developed.
The cool part is related to lipophilic statins were mainly found to be associated with a reduced risk of breast cancer recurrence or mortality particularly on the ER‐negative phenotype. That's original, and it is important to improve the clinical recomandations.
Instead no human studies have explored the effects of ezetimibe, plant sterols, of phytosterols and of fibrateon about breast cancer risk and prognosis.
Therefore, considering the important role of cholesterol in breast cancer development, it is important to clearly specify the different studies on therapies in the conclusions.
Author Response
Reviewer 1
The authors proposed an interesting review. This review is part of a scientific debate on breast cancer risk. The review has involved a detailed and comprehensive plan. The main subject is well developed.
The cool part is related to lipophilic statins were mainly found to be associated with a reduced risk of breast cancer recurrence or mortality particularly on the ER‐negative phenotype. That's original, and it is important to improve the clinical recommendations.
Instead no human studies have explored the effects of ezetimibe, plant sterols, of phytosterols and of fibrateon about breast cancer risk and prognosis.
Therefore, considering the important role of cholesterol in breast cancer development, it is important to clearly specify the different studies on therapies in the conclusions.
We thank the positive report of the reviewer. As requested, we have specified the results in the different therapies in the conclusion section:
“Lipophilic statins seem a good strategy for protecting against breast cancer recurrence and death. However, more studies in humans are necessary to evaluate the role of other therapies, such as ezetimibe, phytosterols or fibrates on breast cancer risk and prognosis.”
Reviewer 2 Report
This review looks at the potential role of HDL and LDL in breast cancer development. It covers the topic at all levels (in vitro, in vivo, and clinical trials). The authors first discuss clinical studies and highlight the fact that there are contradictory results with regards to the significance of LDL/HDL in promoting or protecting against breast cancer. The authors then look in more detail at relevant in vitro/in vivo studies and discuss key pathways (proliferation, apoptosis, etc) that may explain the differences in clinical findings.
Overall well written and a comprehensive review. I only have 2 very minor comments.
Line 253: no space at end of sentence before author number
When consecutive sentences are from the same author they don't all need to be referenced. As an example the sentence from line 251 - 253, the paragraph from lines 177 - 184 all appears to come from ref 55 but referenced 3 times.
Author Response
This review looks at the potential role of HDL and LDL in breast cancer development. It covers the topic at all levels (in vitro, in vivo, and clinical trials). The authors first discuss clinical studies and highlight the fact that there are contradictory results with regards to the significance of LDL/HDL in promoting or protecting against breast cancer. The authors then look in more detail at relevant in vitro/in vivo studies and discuss key pathways (proliferation, apoptosis, etc) that may explain the differences in clinical findings.
Overall well written and a comprehensive review. I only have 2 very minor comments.
Line 253: no space at end of sentence before author number
When consecutive sentences are from the same author they don't all need to be referenced. As an example the sentence from line 251 - 253, the paragraph from lines 177 - 184 all appears to come from ref 55 but referenced 3 times.
We are grateful the Reviewer for his/her comments and suggestions which have helped us to improve the manuscript. As requested, we have added a space in line 253. Moreover, we have reviewed all the manuscript and we have eliminated consecutively repeated references in the same paragraph.
Reviewer 3 Report
This is a nicely written and comprehensive review of articles about the roles of lipid biomarkers in breast cancer development. There are are few suggestion that may help improve the paper quality.
Lines 39-42: Include recent statistics for breast cancer. What is cited in the paper are statistics from 2015.
Mention the search engines and databases used for the review.
Include the criteria (eg. MESH words) used for article selection. You may include a diagram such as a flow chart for clarity.
Were there any inclusion and exclusion criteria for the articles under review? This should be stated in the paper.
Author Response
This is a nicely written and comprehensive review of articles about the roles of lipid biomarkers in breast cancer development. There are few suggestions that may help improve the paper quality.
Lines 39-42: Include recent statistics for breast cancer. What is cited in the paper are statistics from 2015.
We are grateful for your critique and suggestions, which have helped us to significantly improve the manuscript. We have updated the statistics for breast cancer, replacing the cited paper of 2016 by the more recent one, published in 2018.
Mention the search engines and databases used for the review.
As requested, we have specified that Pubmed was the database used in this review.
Include the criteria (eg. MESH words) used for article selection. You may include a diagram such as a flow chart for clarity.
Were there any inclusion and exclusion criteria for the articles under review? This should be stated in the paper.
PubMed was searched comprehensively with combinations of the keyword “breast cancer” and the rest of keywords related with cholesterol and lipoproteins. All relevant papers for the topic of the review were included. This point has been included at the end of Introduction.